# Analysis of Comprehensive Land Consolidation in Cultivated Land Reserve Resource Areas Based on Two-Level Geographical Unit Division

Shaner Li [1], Chao Zhang [1,*], Chenying Luo [1], Shaoshuai Li [2], Wenju Yun [2] and Bingbo Gao [1]

[1] College of Land Science and Technology, China Agricultural University, Beijing 100193, China; lishaner32@cau.edu.cn (S.L.); luochenying57@cau.edu.cn (C.L.); gaobingbo@cau.edu.cn (B.G.)

[2] Land Consolidation and Rehabilitation Center, Ministry of Natural Resources of the People's Republic of China, Beijing 100035, China; lishaoshuai@lcrc.org.cn (S.L.); yunwenju@lcrc.org.cn (W.Y.)

* Correspondence: zhangchaobj@cau.edu.cn

**Abstract:** The scientific and reasonable division of geographical units of cultivated land reserve resource areas is an important foundation for comprehensive land consolidation (CLC). Saline–alkali land is an important cultivated land reserve resource, and is significant for ensuring food security. This paper constructs a two-level land consolidation index system for cultivated land reserve resource areas. The Kruskal graph theory clustering algorithm was used to divide the study area into four types of ecological zones. On the basis of ecological zones, the study area was further divided into three types of consolidation units. Finally, the spatial relationship between ecological zoning and land use types was clarified, and the classification distribution of land consolidation potential was determined through an analysis of the CLC of two-level geographical units. Da'an City, Jilin province, China, was selected as the study area. The results of this study show the following: (1) In Da'an City, the conservation priority areas are concentrated in the north, the development priority areas are concentrated in the central and southern areas, and the comprehensive upgrade areas and adjustment rebuilding areas are in the transition zones. (2) The CLC potential trend is low in the north but high in the south in Da'an City. This paper proposes a framework for a geographical unit delineation method for saline–alkali-enriched cultivated land reserve resource zones, and analyzes the spatial layout of ecological protection demands and land consolidation potential in Da'an City. The results and conclusions of this study will provide a reference for CLC in cultivated land reserve resource areas.

**Keywords:** geographical unit; graph theory clustering; comprehensive land consolidation; cultivated land reserve resources; saline–alkali land

## 1. Introduction

Saline–alkali land is an important reserve resource for cultivated land. According to information released by the Food and Agriculture Organization of the United Nations (FAO) in 2021, the global area of saline–alkali land is close to $8.33 \times 10^8$ ha. It is mainly distributed in natural arid or semi-arid zones in Africa, Asia, and Latin America. In China, the total area of saline–alkali land is about $3.6 \times 10^7$ ha [1]. It is mainly distributed in the north-western, north-central, north-eastern, and coastal regions of China. Unreasonable development and utilization may lead to regional ecological problems such as land sanding, the intensification of secondary salinization, and insufficient water supply. The comprehensive consolidation and development of saline–alkali land is one of the effective ways to guarantee sufficient cultivated land quantity. This is usually achieved through comprehensive land consolidation (CLC) projects. CLC is a systematic project which includes replenishing the quantity of cultivated land, improving the quality of cultivated land, and ecological restoration [2]. How to coordinate the contradiction between CLC and ecological protection has also become a hot topic in current research. CLC projects are

usually carried out in a city or a county. The project area is large and spatially varied. The delineation of geographical units is the basis for formulating scientific and reasonable CLC plans. Therefore, the division of geographical units using scientific methods is of important scientific significance and practical value for promoting the sustainable utilization of regional natural resources and the coordinated development of the ecological environment, and ensuring food security and regional ecological security.

Geographic units are relatively consistent units that are divided by the geographical environment and regional differences. The methods employed in the divided geographical units include remote sensing (RS) recognition [3–7], overlay analysis [8], land type clustering [9–15], etc. RS recognition is rich in data sources. But the geographical unit boundaries do not coincide with land use data and administrative boundaries, so they are not easy to manage. The geographical unit boundaries divided by the overlay analysis are consistent with the land use data and administrative boundaries. However, the overlaying of layers will produce more small units, which is not conducive to CLC. The boundaries of geographical units used in the clustering methods are consistent with the boundaries of current land use data, and the contiguity of map patches is also considered to a certain extent. However, for the clustering method, the spatial distribution characteristics of the area are not considered to be enough, and the result of dividing the units is poor in integrity, which is not conducive to the analysis and planning management of the comprehensive consolidation of saline–alkali land. In order to ensure the spatial continuity of geographical units, the constrained clustering method needs to be improved. In graph theory clustering [16], the internal similarity and external differences of the divided units are ensured, while the integrity of the region is taken into account. This method is usually used in the path planning [17,18], network structure construction [19,20], and medical fields [21]. It was also used to divide geographical space [22] due to its special graph structure.

The division of geographical units is helpful for CLC. But to formulate a scientific and reasonable land consolidation plan, it is also necessary to conduct a potential evaluation of the land consolidation area. There are different priorities and different evaluation methods for CLC in different regions. The evaluation indicators include climate indicators [23], soil indicators [23,24], terrain indicators [23–25], farming conditions [24], economic and demographic indicators [24,25], infrastructure indicators [23–25], and ecological environment indicators [24]. In the existing research, for the evaluation of CLC, improving land use efficiency and improving land productivity are mainly focused on. For cultivated land reserve resource areas, the evaluation of land consolidation potential and ecological environmental impact need to be considered comprehensively. When the evaluation of CLC potential is carried out, it is necessary to combine the development of cultivated land reserve resources and actual needs as appropriate indicators.

Da'an City in Jilin Province, China, which is a commercial grain base of China and an important area for supplementing cultivated land, was selected as the study area in this paper. Ecological land and cultivated land undergo cross-distribution in Da'an City, and saline–alkali land is concentrated and contiguous, but the land resource structure is unreasonable. In this study, the distribution of land resources and ecological environment in Da'an City were clarified, the ecological conditions, resource distribution, and land development potential of different regions were analyzed, and the geographical units at two levels of scales were scientifically divided using the graph theoretic clustering method. For the large-scale units, the coordinating relationship between regional ecological protection and development was considered from a macro perspective, while for the small-scale units, the land development potential was evaluated within the unit from a micro perspective. The relationship between ecological and environmental resources and land development resources in the region was comprehensively analyzed to provide support for land resource allocation adjustment, the CLC plan, and land policy formulation in cultivated land reserve resource areas.

## 2. Materials and Methods

### 2.1. Study Area

Da'an City (Figure 1) is located in the northwest of Jilin Province, in the hinterland of the Songnen Plain, with a total area of 4879 km$^2$. The terrain of Da'an City is low, flat, and open, with small fluctuations, and it is higher in the east and west and lower in the middle. The Nenjiang River, Taoer River, and Huolin River flow through Da'an City. There are also a large number of lakes and marshes distributed in Da'an City, and it is rich in wetlands and surface water resources. Da'an City is one of the three largest soda-type saline–alkali land areas in the world, with large amounts of concentrated and contiguous saline–alkali land. Data from the third national land resource survey of China showed that the saline–alkali land area of Da'an City was 640 km$^2$, accounting for 13.1% of Da'an City's land area.

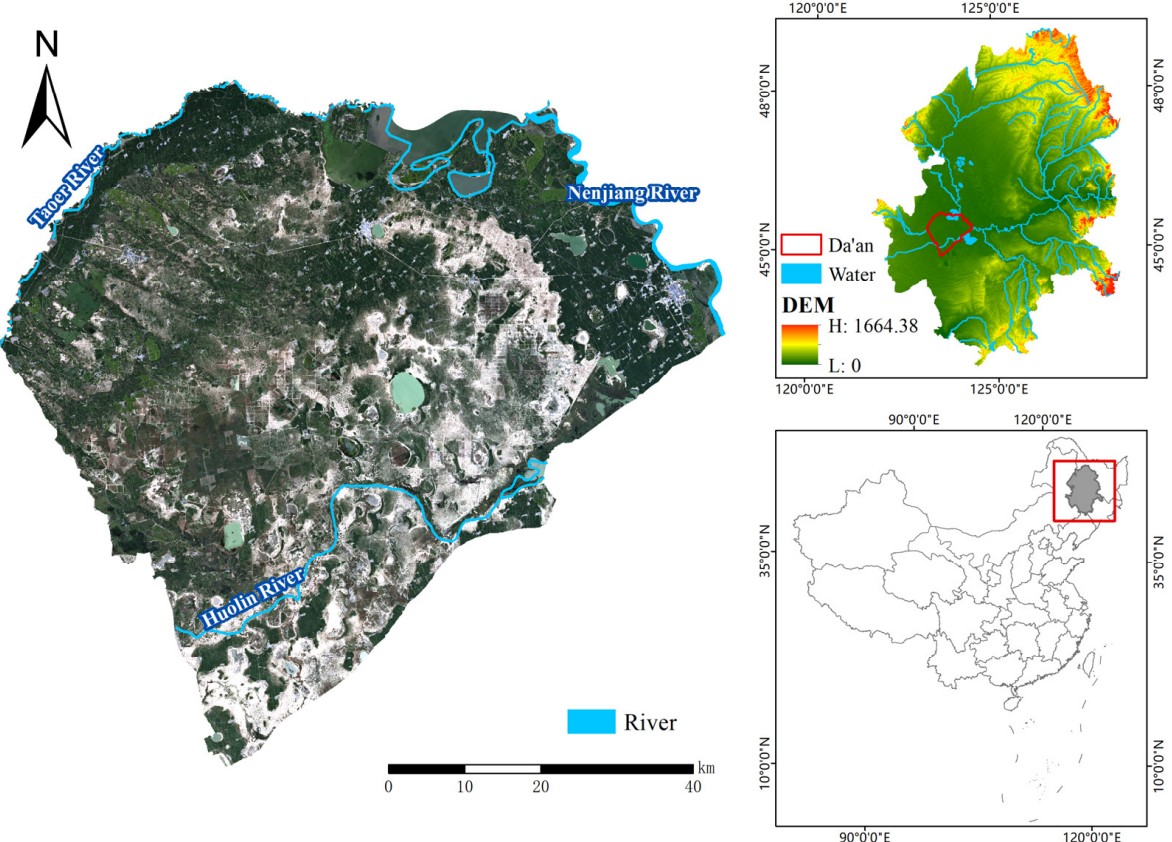

**Figure 1.** Location of Da'an City.

### 2.2. Data Sources and Preprocessing

#### 2.2.1. Data Sources

In this paper, the updated 2021 land use data of Da'an City was obtained from Da'an City Bureau of Natural Resources. The 12.5 m DEM data were obtained from the Japan Aerospace Exploration Institute (https://search.asf.alaska.edu/#/ accessed on 13 January 2022). The 10 m resolution satellite imagery data used for this study consist of the July 2020 Sentinel-2 multispectral remote sensing images obtained from the ESA Copernicus Data Center (https://scihub.copernicus.eu/dhus/#/home accessed on 11 January 2021).

#### 2.2.2. Data Preprocessing

The categorization of land use types in Da'an City is too complicated. In this paper, the land use data were categorized according to the needs of the study, and the control relationship is shown in Table 1.

**Table 1.** Comparison of land use types.

| Land Use Category | Land Use |
|---|---|
| Agricultural land | Cultivated land, orchard, facility agricultural land |
| Ecological land | Forest, grass land, water body, wetland |
| Construction land | Urban construction land, transportation land, Industrial mining and storage land |
| Unutilized land | Saline–alkali land, weed land, bare land, idle land |

*2.3. Methods*

The methodological framework for the delineation of the two-level geographical unit division is shown in Figure 2. (1) The two-level basic unit was divided by multi-scale image segmentation based on the RS data of Da'an City in July 2020 and the ecological red line. (2) Based on the basic unit, an evaluation of CLC in Da'an City was carried out. The calculation results of the evaluation index were used as the attribute value of each basic unit. (3) The two-level geographical unit in Da'an City was divided by graph theory clustering. The Kruskal algorithm [26] and Prim algorithm [27] are the two most common methods in graph theory clustering. In this paper, the Kruskal algorithm is more efficient because the number of basic units is less and the connected graph is sparse. (4) Based on the attributes of basic units, an ecological and developmental collaborative analysis was performed for ecological zones, and land consolidation potential was evaluated for land consolidation units. The type of the two-level geographical unit was determined to complete the two-level geographical unit division of Da'an City.

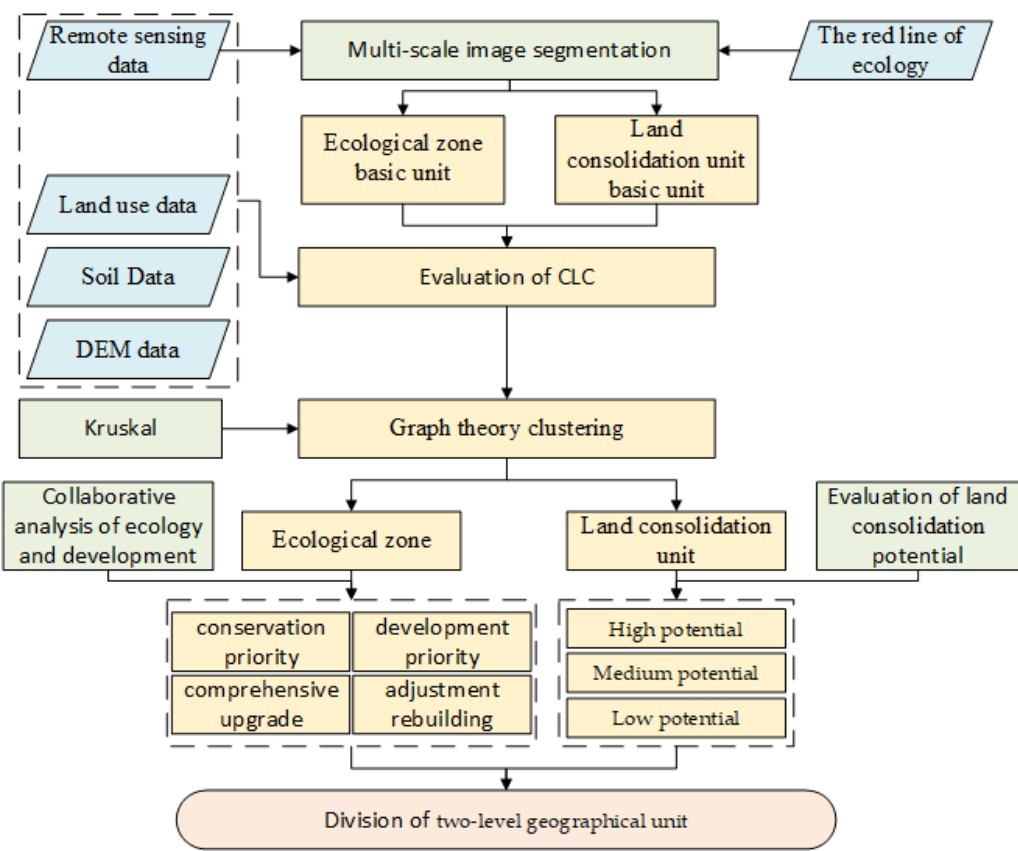

**Figure 2.** Methodological framework of two-level geographical unit division.

2.3.1. Basic Unit Division

Multi-scale image segmentation was used to segment the RS data gathered in July 2020. Multi-scale image segmentation is a method for segmenting images at different scales.

It limits the accuracy of segmented images by setting different segmentation thresholds. The specific steps for determining the basic units were as follows: (1) The basic unit of the first-level ecological zone was segmented. Multi-scale segmentation was performed multiple times on the whole image of Da'an City to determine the threshold value that can segment different land use types. The basic unit of the ecological zone was segmented based on this threshold. (2) The CLC area was identified. The red line of ecology, construction land, pastures, rivers, lakes, reservoirs, and large pits cannot be changed. Areas in Da'an City other than those were considered to reside in the CLC area. (3) The basic unit of the second-level land consolidation unit was segmented. Multi-scale segmentation was performed multiple times on the CLC area of Da'an City to determine the threshold value that can segment different land plots. This threshold was used to segment the basic unit of land consolidation based on the basic unit of ecological zones.

2.3.2. Evaluation of Basic Unit for CLC

In this paper, the degree of land use and the degree of saline–alkali land contiguity were selected as indicators of the development dimension, and the degree of vegetation cover and the value of ecological services per unit area were selected as indicators of the ecological dimension to calculate the attributes of the basic unit of the ecological zone. The proportion of saline–alkali land area, degree of soil salinization, distance from the water source, and average slope were selected as indicators of the land consolidation potential dimension, and the attributes of the basic land consolidation units were calculated. The indicators and calculation methods are shown in Table 2.

**Table 2.** Indicator system and weight.

| Level | Dimension | Index | Formula | Relativity | Weight |
|-------|-----------|-------|---------|:----------:|:------:|
| Ecological zone | Development | Land use degree [1] | $100 \times \sum_{i=1}^{n} A_i \times C_i$ | − | 0.33 |
| | | Saline–alkali land connectivity | *Saline–alkali land area / number of saline–alkali land areas* | + | 0.67 |
| | Ecology | Vegetation coverage | $(NDVI - NDVImin)/(NDVImax - NDVImin)$ | + | 0.28 |
| | | Ecological service value [2] | $(\sum S_i \times VC_i)/area$ | + | 0.72 |
| Land consolidation unit | Land potential | Proportion of saline–alkali land | *Saline–alkali land area / area × 100%* | + | 0.49 |
| | | Soil salinity degree | | − | 0.17 |
| | | Distance from water source | $\sqrt{(x_1 - x_2)^2 + (y_1 - y_2)^2}$ | + | 0.22 |
| | | Slope | | − | 0.12 |

[1] The land use grading index in the table refers to the four land use grades under ideal conditions proposed by Zhuang Dafang et al. [28]. [2] The ecological service value per unit area was calculated with the method proposed by Costanza et al. [29].

CLC has been affected by a variety of factors in nature. In order to reflect the differences in the degree of importance between different influence factors, weights need to be assigned to the indicators when calculating the basic unit attributes. The entropy weight method is an objective weighting method based on information entropy that reduces the influence of subjective factors on the weight of indicators. The formula for calculating the weights of n units of m indicators is as follows:

$$W_j = \frac{1 - E_j}{m - \sum E_j} \tag{1}$$

$W_j$ is the weight of the index $j$ and $E_j$ is the entropy of the index $j$. The formula is as follows:

$$E_j = -\frac{\sum_{i=1}^{n} p_{ij} \times \ln p_{ij}}{\ln n} \tag{2}$$

$$p_{ij} = \frac{Y_{ij}}{\sum_{i=1}^{n} Y_{ij}} \tag{3}$$

$Y_{ij}$ is the standardized value of the index $j$ in the region $i$.

### 2.3.3. Graph Theory Clustering Based on Kruskal Algorithm

(1)    Connected graph construction

In the study of geographical unit delineation, the consistency within the unit and differentiation from the other units is ensured, and the spatial connectivity and completeness of the unit is satisfied through graph theory clustering. The basis for conducting graph theory clustering is the construction of a connected graph. A connected graph is a collection of connections between vertices in space, denoted as G = {V, E, D}, where V represents the set of vertices, E denotes the set of edges, and D represents the weights of the edges. The connected graph is constructed in the area to be divided. Firstly, the basic units are abstracted into a number of vertices. Then, the adjacency between basic units is represented by the edges of the connected graph. Finally, the Euclidean distance between two neighboring basic units of each indicator dimension is calculated based on the weights of the edges. The calculation formula is as follows:

$$D_{ij} = \sqrt{\sum_{k=1}^{3} \left( x_{ik} - x_{jk} \right)^2} \tag{4}$$

$D_{ij}$ is the weight of the edge between evaluation units $i$ and $j$; $x_{ik}$ and $x_{jk}$ are the index $k$ of evaluation units $i$ and $j$, respectively.

(2)    Principle of Kruskal algorithm

A minimum spanning tree (MST) is a minimal connected subgraph that contains all the vertices in the connected graph whereby the sum of the weights of the edges is minimized. There are n-1 edges in an MST with n vertices, and there are no loop paths in the tree. The basic idea of the Kruskal algorithm is to find the edge with the smallest weight in the connected graph to add to the tree, and there are no loop paths in the tree after adding the edge. First of all, the connected graph is sorted in accordance with the weight of the edge from small to large. Then, the edge with the smallest weight is selected. If there is no loop path in the tree after adding the edge, the edge is added to the tree; otherwise, it is discarded. Finally, the MST is formed until the number of edges in the tree is less than the number of vertices in the original connected graph by one. The steps of the Kruskal algorithm are shown in Figure 3.

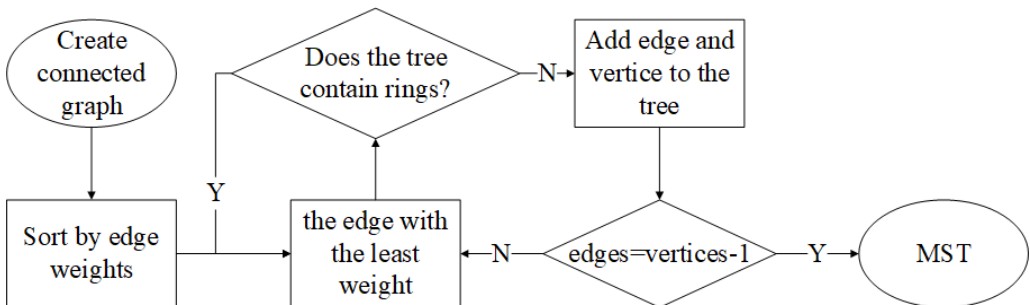

**Figure 3.** Kruskal algorithm flowchart.

(3)    Implementation of graph theory clustering

The connection of the smallest differences between the basic units in the region is represented by the MST. At this point, all basic units still belong to the same partition. The MST needs to be divided to form multiple partitions. There are two methods of division:

(1) According to the weights of the edges in the tree, a certain threshold is set. Edges with weights greater than this threshold are discarded, and the region is divided into several partitions. This method divides the units by cutting off the connection between the basic units with large differences. (2) According to the structure of the MST, the appropriate sub-node branches are selected for partitioning; thus, several sub-trees are formed as several units in the study area. This method divides the units by severing the links between the groups of basic units with large differences. There are wetlands such as Nenjiang Mudflats, Chagan Lake Swamp, and Moon-lake reservoir in Da'an City. There are significant differences between these wetlands and other land types. If they are divided according to the first method, the wetlands will form a partition of their own, and the other basic units will form a large partition. Such a result is not conducive to the regional CLC. Therefore, the method of dividing units on the branching partition of the MST was chosen in this paper. With this approach, internal consistency and external differentiation of the unit were ensured, and the actual demands for CLC were met.

## 3. Results

### 3.1. Ecological Zone

3.1.1. Ecological Zone Distribution

The RS image of Da'an City was segmented to obtain 400 basic ecological units. A connected graph, with the basic units as vertices, the neighborhoods of the basic units as edges, and the values of ecological and developmental attributes as the weights of the edges, was constructed. The Kruskal algorithm was used to generate an MST, and according to the structure of the MST, the branch partitioning method was used to divide the ecological zones. The basic units on the same branch of the tree were divided into the same partition. The division is shown in Figure 4. Finally, 20 ecological zones were divided in the whole area of Da'an City and they were numbered with A-T, as shown in Figure 5.

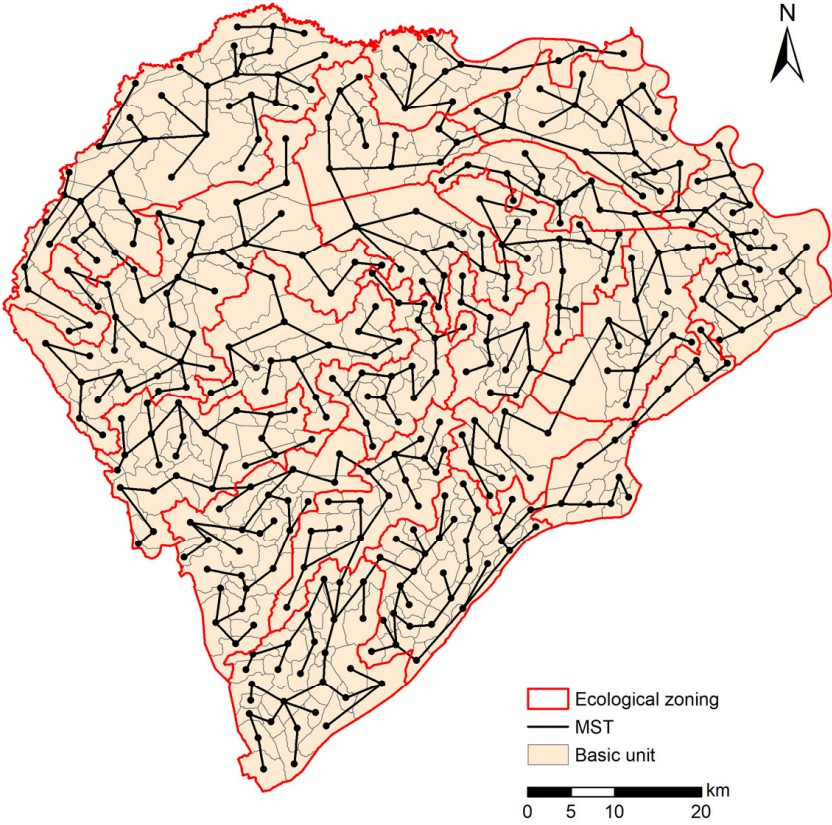

**Figure 4.** MST division of ecological zones.

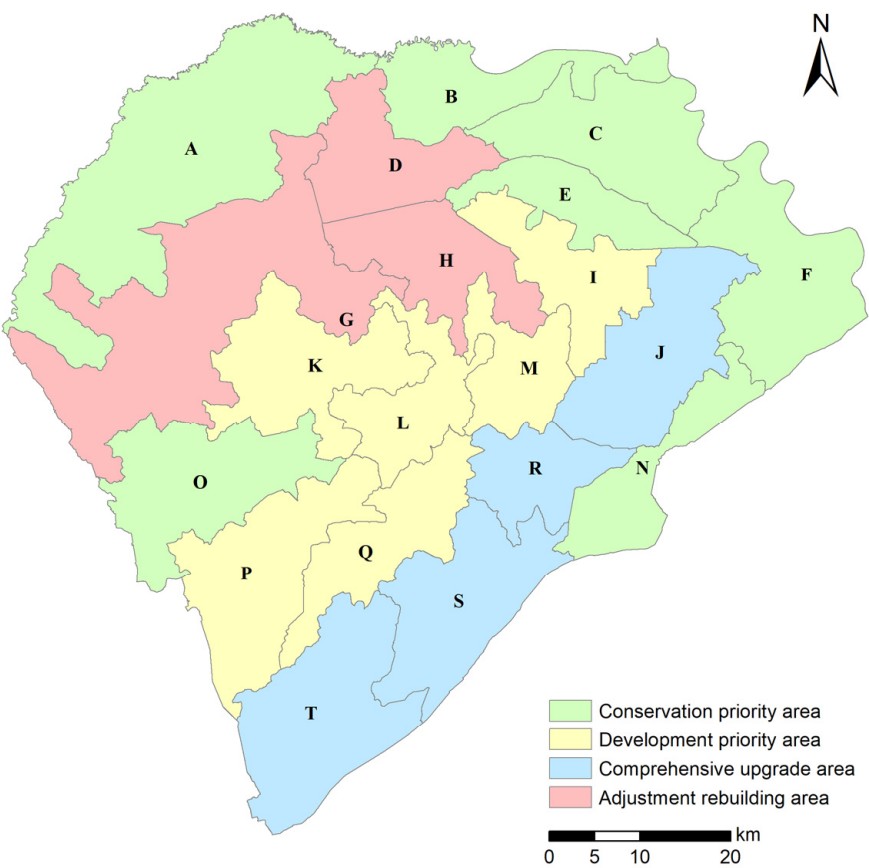

**Figure 5.** Distribution of ecological zone types.

3.1.2. Ecological Zone Type

After determining the spatial distribution of ecological zones by segmenting the MST with graph theory clustering, the types of zones were determined based on their attribute values. The type of basic unit was determined by analyzing the synergistic relationship between the indicators of the ecological and developmental attributes of the basic units. And then, the type of zone is determined by calculating the area proportion of each type of basic unit within the zone. A two-dimensional coordinate system was constructed with development as the horizontal coordinate and ecology as the vertical coordinate. And all basic units were partitioned into four quadrants to form four types of basic units, as shown in Figure 6. The basic unit for which both development and ecological protection are located in the high-value area (first quadrant), is identified as the comprehensive upgrade area. In the comprehensive upgrade area, both the requirement of ecological conservation and agricultural production are important. They need to be integrated with development. The basic unit whose development is located in the low-value area but whose ecological protection is located in the high-value area (second quadrant) is identified as the conservation priority area. Ecological condition is prominent in the conservation priority area, where ecological functions outweigh development requirements. The basic unit for which both development and ecological protection are in located low-value areas (third quadrant) is identified as an adjustment rebuilding area. Ecological conditions and development potential are poor in the adjustment rebuilding area. Its adjustment is urgently needed. The basic unit whose development is in the high-value area and ecological protection in the low-value area (fourth quadrant) is identified as a development priority area. Development potential is prominent in the development priority area, where development requirements outweigh ecological functions.

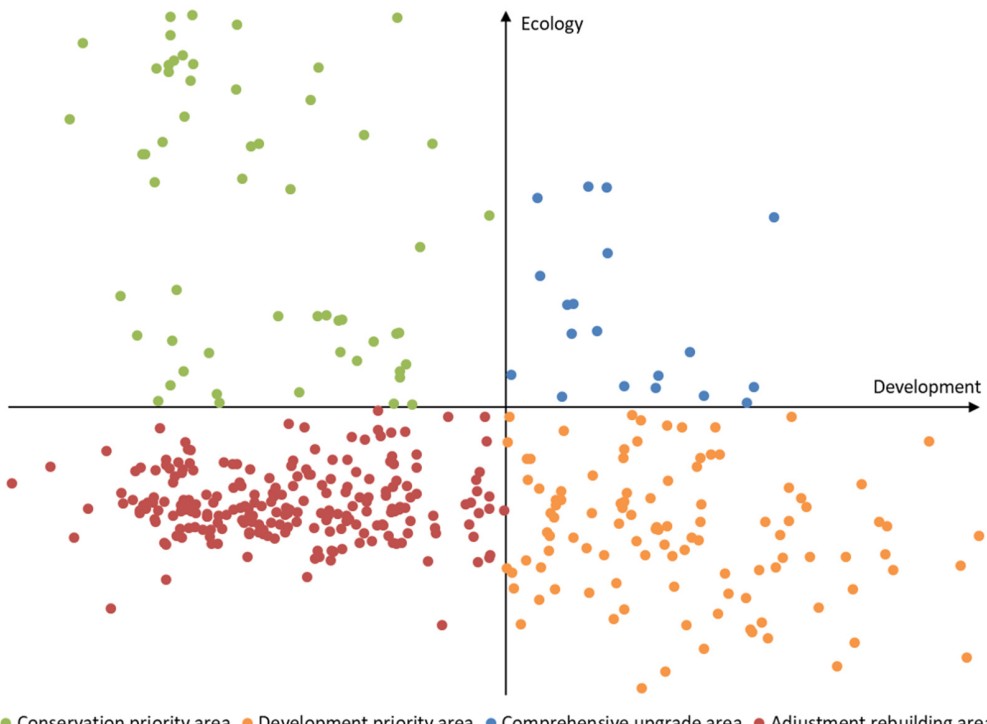

Figure 6. Synergistic relationship between development and ecological protection.

The types of the twenty ecological zones are shown in Table 3. There are seven conservation priority areas, with a total area of 1824.03 km², accounting for 37.39% of the total area, mainly distributed in the northern and eastern regions and part of the western region. These areas are concentrated distribution areas of wetlands and other ecological land in Da'an City. The Nenjiang River and Tao'er River are in the northern area, Chagan Lake is in the southeast area, and the Niuxintoubao National Wetland Park is in the western area in Da'an City. There are six development priority areas, with a total area of 1177.47 km², accounting for 24.13% of the total area, distributed in the central and south-central areas of Da'an City. Unutilized land such as saline–alkali land and weed land in Da'an City is concentrated in this area. There are four comprehensive upgrade areas, with a total area of 937.66 km², accounting for 19.22% of the whole area, distributed in the transition area between the conservation priority area and development priority area in the east and south of Da'an City. Saline–alkali land is also more widely distributed in this area, but CLC projects have been carried out over a larger area. There are three adjustment rebuilding areas, with a total area of 939.87 km², accounting for 19.26% of the total area, distributed in the transition area between the conservation priority area and development priority area in the northeast of Da'an City, which is mostly a cross-distributed area of forest land, cultivated land, and construction land.

Table 3. Basic unit types and areas of ecological zones (unit: km²).

| Ecological Zone | Conservation Priority Area | Adjustment Rebuilding Area | Development Priority Area | Comprehensive Upgrade Area | Total Area | Type |
|---|---|---|---|---|---|---|
| A | 287.55 | 232.22 | 46.26 | 9.80 | 575.82 | Conservation priority area |
| B | 178.88 | 0.00 | 0.00 | 0.00 | 178.88 | Conservation priority area |
| C | 148.38 | 34.91 | 0.00 | 83.59 | 266.88 | Conservation priority area |
| D | 0.00 | 94.01 | 79.06 | 12.18 | 185.25 | Adjustment rebuilding area |

**Table 3.** *Cont.*

| Ecological Zone | Conservation Priority Area | Adjustment Rebuilding Area | Development Priority Area | Comprehensive Upgrade Area | Total Area | Type |
|---|---|---|---|---|---|---|
| E | 77.79 | 4.41 | 0.00 | 45.25 | 127.44 | Conservation priority area |
| F | 121.08 | 116.13 | 0.00 | 31.03 | 268.24 | Conservation priority area |
| G | 50.73 | 425.16 | 52.98 | 32.37 | 561.25 | Adjustment rebuilding area |
| H | 0.00 | 162.78 | 22.13 | 8.46 | 193.37 | Adjustment rebuilding area |
| I | 42.43 | 12.92 | 86.12 | 33.08 | 174.55 | Development priority area |
| J | 31.54 | 3.63 | 36.14 | 182.26 | 253.57 | Comprehensive upgrade area |
| K | 0.00 | 0.00 | 236.15 | 10.99 | 247.15 | Development priority area |
| L | 26.02 | 54.00 | 65.52 | 10.92 | 156.46 | Development priority area |
| M | 0.00 | 5.20 | 100.99 | 27.99 | 134.18 | Development priority area |
| N | 72.60 | 6.77 | 11.79 | 57.83 | 148.99 | Conservation priority area |
| O | 104.08 | 12.33 | 54.80 | 86.56 | 257.76 | Conservation priority area |
| P | 59.14 | 17.57 | 109.76 | 88.61 | 275.08 | Development priority area |
| Q | 0.00 | 0.00 | 107.84 | 82.22 | 190.06 | Development priority area |
| R | 10.59 | 14.24 | 33.11 | 74.04 | 131.98 | Comprehensive upgrade area |
| S | 15.08 | 7.73 | 105.36 | 122.34 | 250.51 | Comprehensive upgrade area |
| T | 29.40 | 19.39 | 107.16 | 145.65 | 301.60 | Comprehensive upgrade area |

### 3.1.3. Land Use Structure of Ecological Zones

The land use structure of the ecological zones was analyzed in conjunction with ecological zone boundaries and land use types in Da'an City. In all four ecological zones of Da'an City, the area proportion of cultivated land is the highest, especially in the conservation priority area and adjustment rebuilding area, where the areas of cultivated land are 945.84 km$^2$ and 601.47 km$^2$, accounting for 51.85% and 63.99% of the total area, respectively. In the conservation priority area, the proportion of the areas of water bodies, wetland, forest, and grass land are significantly higher than for other types of land, at 11.78%, 9.43%, 8.46%, and 6.41%, respectively. The sum of the area of construction land and unutilized land is only 11.86% of the total area. In the development priority area, the area of unutilized land accounts for 47.59% of the total region area. Among the unutilized land, the most abundant land is saline–alkali land and weed land, accounting for 24.22% and 23.22% of the total area of the region, respectively. The distribution of land use types in the comprehensive upgrade area is similar to that in the development priority area. There is slightly more ecological land such as forest and grass land than in the development priority area, and less unutilized land such as saline–alkali land and weed land than in the development priority area. Different from the development priority area, most of the cultivated land in this area is formed of land improvement projects and saline–alkali land development and treatment projects, which are distributed in a more concentrated and contiguous manner. In the adjustment rebuilding area, cultivated land, forest land, weed land, and construction land are staggered, with area proportions of 63.99%, 14.26%, 7.74%,

and 4.86%, respectively. The distribution of land use types in the ecological zones is shown in Figure 7, and the area and proportion of land use types in each type of zone are shown in Table 4.

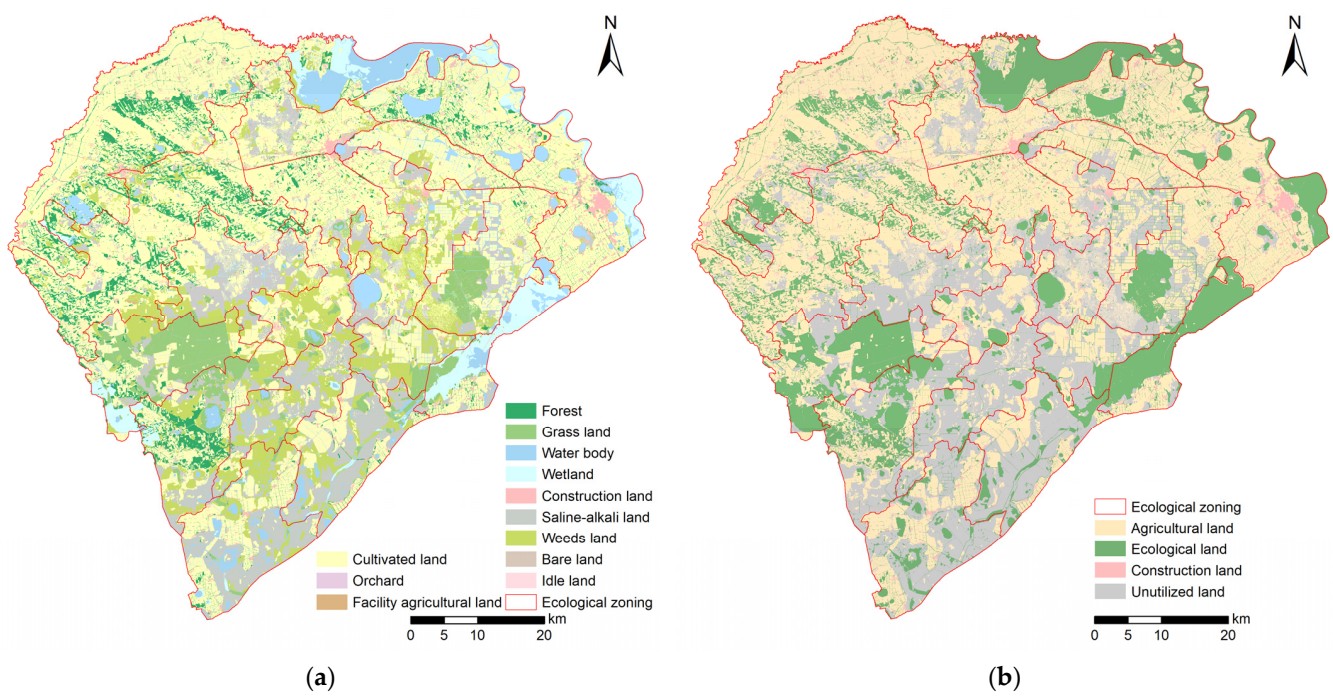

**Figure 7.** (**a**) Distribution of land use types in ecological zones; (**b**) distribution of land use category in ecological zones.

**Table 4.** Areas of land use types in various ecological zones.

| Land Use Type | Conservation Priority Area | | Development Priority Area | | Comprehensive Upgrade Area | | Adjustment Rebuilding Area | |
|---|---|---|---|---|---|---|---|---|
| | Area/km² | Proportion | Area/km² | Proportion | Area/km² | Proportion | Area/km² | Proportion |
| Cultivated land | 945.84 | 51.85% | 397.08 | 33.72% | 322.66 | 34.41% | 601.47 | 63.99% |
| Orchard | 1.28 | 0.07% | 0.03 | 0.00% | 0.06 | 0.01% | 0.13 | 0.01% |
| Facility agricultural land | 2.20 | 0.12% | 0.62 | 0.05% | 0.86 | 0.09% | 1.59 | 0.17% |
| Forest | 154.35 | 8.46% | 61.91 | 5.26% | 16.92 | 1.80% | 134.05 | 14.26% |
| Grass land | 117.00 | 6.41% | 57.39 | 4.87% | 78.20 | 8.34% | 5.24 | 0.56% |
| Water body | 214.87 | 11.78% | 63.08 | 5.36% | 74.59 | 7.95% | 12.35 | 1.31% |
| Wetland | 172.09 | 9.43% | 8.76 | 0.74% | 10.16 | 1.08% | 2.86 | 0.30% |
| Construction land | 82.27 | 4.51% | 28.27 | 2.40% | 25.35 | 2.70% | 45.71 | 4.86% |
| Saline–alkali land | 50.07 | 2.75% | 285.22 | 24.22% | 241.78 | 25.79% | 63.23 | 6.73% |
| Weed land | 83.10 | 4.56% | 273.40 | 23.22% | 166.08 | 17.71% | 72.72 | 7.74% |
| Bare land | 0.94 | 0.05% | 1.71 | 0.15% | 1.00 | 0.11% | 0.49 | 0.05% |
| Idle land | 0.01 | 0.00% | 0.00 | 0.00% | 0.00 | 0.00% | 0.03 | 0.00% |

The spatial superposition analysis of the ecological zones and land use types shows that ecological land such as forest, grass land, water bodies, and wetland are mainly distributed in conservation priority areas, and land use types that can be subject to land consolidation, such as saline–alkali land and weed land, are mainly distributed in the development priority areas. So, the division of ecological zones and the determination of land types in Da'an City are scientific and reasonable.

### 3.2. Land Consolidation Units

3.2.1. Distribution of Land Consolidation Units

Based on the ecological zone segmentation, the RS image of Da'an City was segmented. After removing the non-development areas, a total of 1412 basic units of land consolidation were obtained. Since part of the area of Da'an City was removed in the process of dividing the land consolidation units, there is a discontinuous distribution of the basic units. Since each zone was divided by the Kruskal algorithm graph theory clustering, it is necessary to discuss the different cases separately. Case 1: A single basic unit is discretely distributed, and there are no adjacent basic units. The basic unit is regarded as a land consolidation unit. Case 2: Fewer than or equal to three basic units are distributed continuously. The MST cannot be divided, and it is meaningless to construct its MST. Therefore, all neighboring basic units are fused into one land consolidation unit. Case 3: There are greater than three basic units continuously distributed. This situation is most common in the study area, and the method of partitioning the MST is used to divide the land consolidation unit. The MST for each zone is shown in Figure 8.

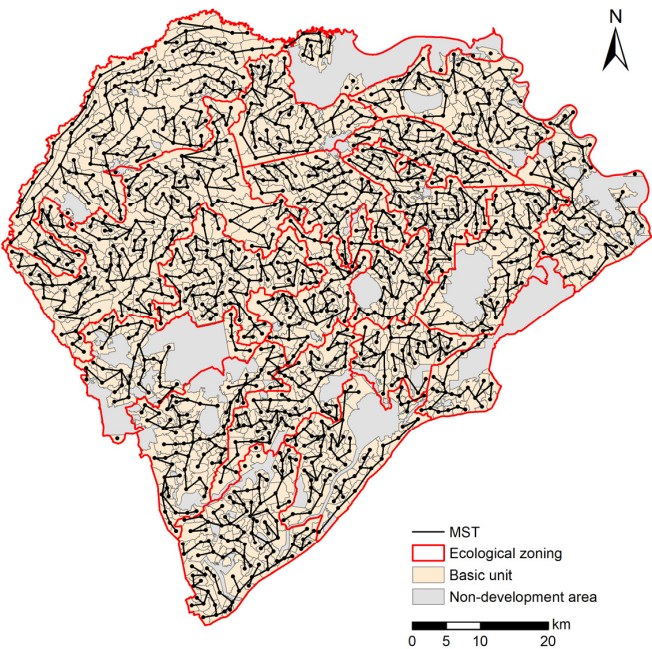

**Figure 8.** MST of land consolidation unit.

3.2.2. Land Consolidation Unit Type

A total of 116 land consolidation units were segmented by dividing the MST. The land potential attribute values of all basic units were calculated. Then, the land consolidation potential of the land consolidation unit was calculated with the area weighting method. The land consolidation potentials were graded from high to low and categorized into three types: high-potential units, medium-potential units, and low-potential units. In this paper, the geometric interval grading method was used to grade land consolidation units. Compared with the natural breakpoint method and the equal interval method, the geometric interval method is advantageous in processing continuous data, with which a balance between the variation in intermediate values and the variation in extreme values is achieved. Then, the range of each level and the number of elements contained in each level are approximately the same.

The land consolidation potential of land consolidation units was graded, and 37 high-potential units, 53 medium-potential units, and 26 low-potential units were obtained. The area of each type of unit is shown in Table 5, and the distribution is shown in Figure 9. The medium-potential units are the most numerous, accounting for 41.04% of the total area, and are mainly located in the adjustment rebuilding area in the northeastern part of

Da'an City. The development priority area is located in the central part. A small number of them are located in the comprehensive upgrade area. High-potential units are the next most abundant, accounting for 26.25% of the total area. They are concentrated in the development priority area in the central part of Da'an City, and the comprehensive upgrade area in the south. Low-potential units are the least numerous, accounting for only 17.33% of the total area. They are distributed in the conservation priority area in the northern part of Da'an City.

**Table 5.** Land consolidation unit types and areas.

| Type of Units | Number of Units | Area/km$^2$ | Proportion |
| --- | --- | --- | --- |
| High-potential unit | 37 | 1280.68 | 26.25% |
| Medium-potential unit | 53 | 2002.13 | 41.04% |
| Low-potential unit | 26 | 845.62 | 17.33% |
| Non-development area | - | 750.60 | 15.38% |
| Total | 116 | 4879.03 | 100% |

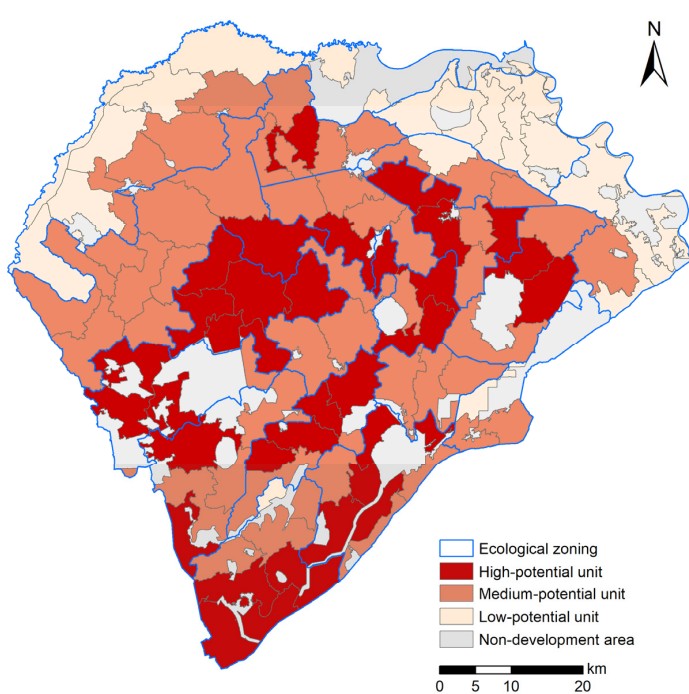

**Figure 9.** Distribution of land consolidation unit types.

### 3.2.3. Potential Analysis of Land Consolidation Unit

CLC in cultivated land reserve resource areas involves the management and utilization of unutilized land. The amount of unutilized land is an important indicator of land consolidation potential. The unutilized land in Da'an City is mainly distributed in the central and southern parts of the city. The distribution of unutilized land is spatially consistent with the distribution of high-potential units and medium-potential units. In the high potential units, the area of unutilized land is 659.01 km$^2$, accounting for 51.46% of the total area of the units. In these units, the developable area is large and the land consolidation potential is high. In medium-potential units, the area of unutilized land is 441.48 km$^2$, accounting for 22.05% of the total area of the units. The land in these units has some development potential. In low-potential units, the area of unutilized land is 39.48 km$^2$, accounting for 4.67% of the total area of the units. In these units, the developable area is much smaller and the land consolidation potential unit is extremely low. The area of unutilized land in each type of unit is shown in Table 6, and the distribution is shown in Figure 10.

**Table 6.** Unutilized land area of land consolidation unit.

| Types of Units | Area/km$^2$ | Proportion |
| --- | --- | --- |
| High-potential unit | 659.01 | 51.46% |
| Medium-potential unit | 441.48 | 22.05% |
| Low-potential unit | 39.48 | 4.67% |

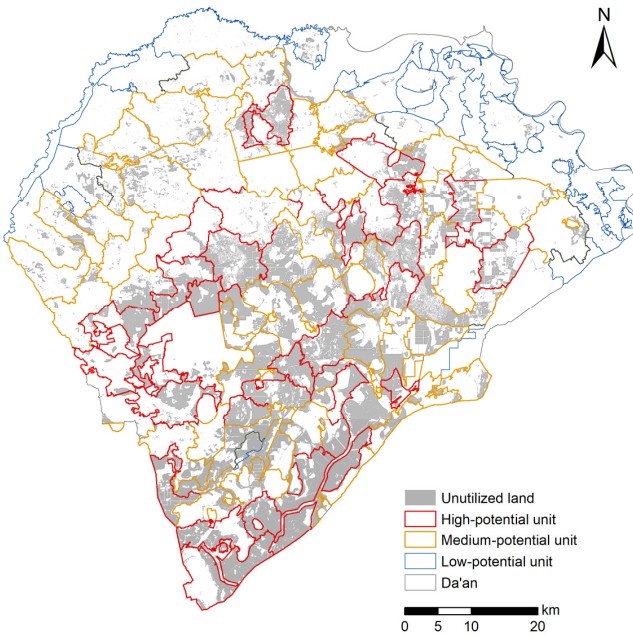

**Figure 10.** Distribution of unutilized land in land consolidation unit.

Da'an City is currently carrying out CLC. The three areas that have been planned are shown in Figure 11. Through the overlay analysis of the land consolidation units and Da'an City's CLC planning areas, it can be seen that all of the planning areas are located within the high-potential units and medium-potential units. Therefore, the results of the division of land consolidation units are in line with the actual situation in Da'an City.

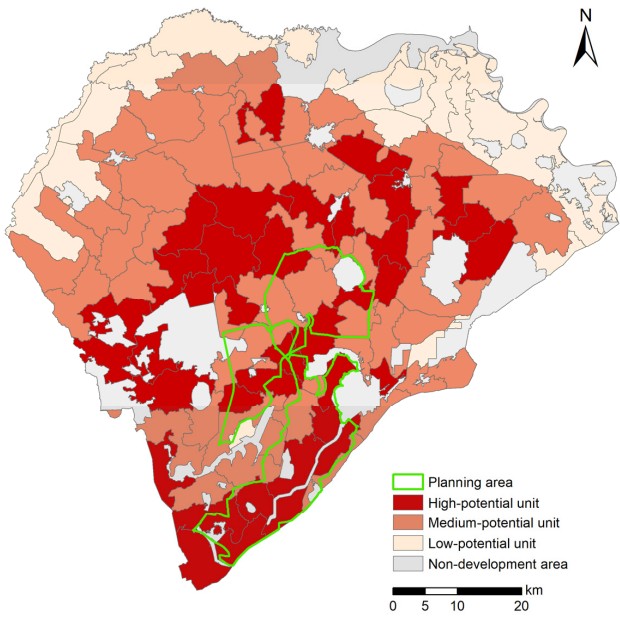

**Figure 11.** Spatial distribution of land consolidation planning area and land consolidation unit.

### 4. Discussion

(1) Diversity of factors influencing CLC

CLC is a systematic project that aims to promote the comprehensive improvement of fields, water, roads, forests, and villages [30]. Topography, soil conditions, the ecological environment, and economic and social factors are the main factors which differently affect the CLC in different areas. For example, in mountainous areas, the main factors affecting CLC are topography, transportation, and economy [31]. In the areas of cultivated land fragmentation, the main considerations are the morphology of cultivated land, spatial distribution, and economic and social factors [32]. The study area of this paper is a cultivated land reserve resource area with saline–alkali land enrichment, the factors with more significant influences on saline–alkali land improvement are shown in Table 2.

(2) Universality of the application of CLC analysis

CLC is one of the important means to coordinate the orderly development of land resources [33]. The analysis of CLC can provide a scientific basis for the spatial planning of regional land. According to the different requirements of different regions, the emphasis of CLC analysis is also different. Luo et al. [33] conducted an analysis of productivity, habitability, and sustainability for urbanization and industrialization. Feng et al. [34] carried out a quantitative analysis of the effects of ecological security on the construction of ecological civilization. Zang et al. [35] conducted an analysis of rural sustainability, land fragmentation, and land redistribution for rural revitalization. This paper focuses on land consolidation, development, and utilization in cultivated land reserve resource areas, and analyzes the CLC according to the synergic relationship between regional land consolidation potential and regional ecological protection.

(3) RS image segmentation combined with graph theory clustering

In studies related to unit division, researchers usually use administrative regions [36,37] or grids [38] as the basic unit. However, the natural geographic boundaries are broken with such a basic unit. It is not conducive to the analysis of the process of CLC. In this paper, RS image segmentation is used to obtain basic units, with which the destruction of natural geographic boundaries is avoided to a large extent. In the method of unit division, the multi-indicator superposition division [39–41] and synergistic relationship division of different indicators are more common [42]. The units divided by these kinds of methods are mostly in a discrete and broken state, and the work of CLC is difficult with these kinds of units. On this basis, the method of clustering is used to aggregate the units. For example, Xiao et al. [38] used self-organizing maps and hierarchical clustering, and Zhao et al. [43] used a combination of a self-organizing feature mapping neural network and fuzzy mean clustering to divide units. The discrete distribution of the units was reduced to a large extent, but the spatial distribution of the basic units was still taken into account with the clustering method. In this paper, both the natural geographic boundaries of the study area and the spatial relationships of the basic units were taken into account with the method of image segmentation combined with graph theory clustering. A centralized distribution was presented in the final partition. The foundation for CLC was constructed with the two-level geographical unit divided in the study area. And the division units obtained with this method are consistent with the natural geographic boundaries, and the spatial continuity is ensured.

(4) Contributions and Limitations

As one of the major cultivated land reserve resource areas in China, developing saline–alkali land and supplementary cultivated land in the study area is important to guarantee national food security. At the same time, there are large areas of ecological land such as wetlands in Da'an City. Relieving the structural contradiction between productive land and ecological protected land in the region is an urgent task for the integrated utilization of land resources. In this paper, the graph theory clustering method is applied to the unit division of cultivated land reserve resource areas. On the basis of studying the distribution

of land resources and functional zoning, this paper scientifically delineates ecological zones and quantifies their land consolidation potential in terms of land consolidation and development. In the process of CLC projects in Da'an City, it provides scientific support for the comprehensive coordination of the management and utilization of saline–alkali land and ecological protection, the scientific formulation of saline–alkali land management and utilization programs, and the achievement of regional sustainable development.

A distinctive feature in Da'an City is the wide distribution of saline–alkali land; the rational development and utilization of saline–alkali land should be focused on. In the process of dividing the geographical units, key factors of saline–alkali land consolidation and utilization, such as saline–alkali land area, soil salinization, and water resources, were used as the main indicators. These indicators are not necessarily related to the delineation of all geographical units of cultivated land reserve resource areas. In other cultivated land reserve resource areas, such as Xinjiang, Gansu, Henan, etc., if this geographical unit division method is adopted, the current situation of local land resources can be considered, and matching indicators should be selected to calculate the attributes of the two-level basic units. Considering the easy access to data, RS data with 10 m resolution were used in this paper. Some information may have been missed in the segmentation of basic units. It remains to be further studied how to carry out a more detailed unit division with higher-resolution data.

## 5. Conclusions

In order to cope with the impact of regional spatial differences on CLC, this paper proposes a two-level geographical unit division framework, which includes RS image segmentation, the evaluation of CLC, and graph theory clustering. The spatial distribution characteristics of geographical units in Da'an City were found by analyzing the spatial distribution of geographical unit types at two levels. The northern part of the city has low land consolidation potential and a high need for ecological protection, which should be emphasized. The central region, with its high land consolidation potential and high demand for land consolidation and development, should be prioritized for land development planning. Both CLC and ecological conservation are important in the southern region, and they need to be upgraded in an integrated manner.

CLC requires spatial unitization as a basis. The framework constructed in this paper is a spatial demarcation method for CLC based on cultivated land reserve resource areas enriched by saline–alkali land. This methodological framework has the potential to be extended to a wider range of applications by simply adapting the evaluation system to suit the study area. A more extensive division of spatial units by graph theory clustering could provide more decision makers with scientific support for CLC.

**Author Contributions:** Conceptualization, S.L. (Shaner Li) and C.Z.; methodology, S.L. (Shaner Li); software, S.L. (Shaner Li); validation, C.L.; formal analysis, S.L. (Shaner Li); investigation, S.L. (Shaner Li); resources, C.Z. and S.L. (Shaoshuai Li); data curation, S.L. (Shaner Li) and C.L.; writing—original draft preparation, S.L. (Shaner Li); writing—review and editing, C.Z., C.L., S.L. (Shaoshuai Li), W.Y., and B.G.; visualization, S.L. (Shaner Li); supervision, W.Y. and S.L. (Shaoshuai Li); project administration, C.Z.; funding acquisition, C.Z. All authors have read and agreed to the published version of the manuscript.

**Funding:** This research was funded by Mechanisms and Processes for The Formation of Key Evaluation Indicators for Different Regional Profiles, National Key Research and Development Program of China, grant number 2021YFD1500202.

**Data Availability Statement:** The updated 2021 land use data of Da'an City were obtained from the Da'an Bureau of Natural Resources. The data are not publicly available due to national security policies. The 12.5m DEM data were obtained from the Japan Aerospace Exploration Institute (https://search.asf.alaska.edu/#/ accessed on 13 January 2022). The 10m resolution Sentinel-2 multispectral RS image data of July 2020 were obtained from the ESA Copernicus Data Center (https://scihub.copernicus.eu/dhus/#/home accessed on 11 January 2021).

**Conflicts of Interest:** The authors declare no conflicts of interest.

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
