# Peer review of "Analysis of Comprehensive Land Consolidation in Cultivated Land Reserve Resource Areas Based on Two-Level Geographical Unit Division"

_land, doi:10.3390/land13040470_

Round 1

Reviewer 1 Report

Comments and Suggestions for Authors

1. It is suggested to supplement the global and Chinese saline-alkali land distribution in the introduction.

2. In the second paragraph of the introduction, it is suggested to briefly summarize the applicable scenarios or advantages and disadvantages of remote sensing identification and superposition analysis.

3. In the last paragraph of the introduction, it is suggested to explain why Da 'an City is suitable for Kruskal algorithm.

4. 3.1.2, the fourth quadrant should be the development priority area.

5. 3.1.2, it is suggested to explain the characteristics of each quadrant.

6. In the discussion, it is suggested to supplement the relevant content of comprehensive land consolidation analysis.

7. In the conclusion, it is suggested to give corresponding suggestions for different partitions.

8. The conclusion is thin, and it is suggested to supplement some.

9. The statement of the conclusion in the abstract does not correspond to the conclusion.

Reviewer 2 Report

Comments and Suggestions for Authors

Dear authors,

Thanks for sharing your work, I have thoroughly read the manuscript and the comments are provided below.

1. The term "saline-alkali land" is included in the keywords, however, its significance is not discussed in the abstract.

2. Line 83 fails to clarify the difference between comprehensive land consolidation implemented within cultivated land reserve resource areas and other geographical locations.

3. The satellite imagery data used for this study consists of July 2020 Sentinel-2 multispectral remote sensing images obtained from ESA Copernicus Data Center (https://scihub.copernicus.eu/dhus/#/home).

4. Does Figure 2 represent verification results for assessing land consolidation potential?

5. The authors may want to improve the DPI of Figure 3.

6. In section 2.3.3 (3), it is recommended to provide an initial explanation of both segmentation methods followed by a discussion on their respective selection basis.

7. It is suggested that distinct colors be utilized for representing ecological zones in Figure 4(b), or alternatively, consider replacing Figure 4(b) with Figure.

8. The title of figure10 does not specify the nature/content depicted on the map.

9. The conclusion section should emphasize the study methods.

10. The thematic map requires the inclusion of a legend.

11. The references 32 and 40 are duplicated.

Comments on the Quality of English Language

Minor editing of English language required.

Reviewer 3 Report

Comments and Suggestions for Authors

This article describes the construction of a two-level land consolidation index system for the cultivated land reserve resources area. Kruskal graph theory clustering algorithm was used to divide the study area into four types of ecological zones.An interesting study for land consolidation projects. Some changes need to be made in order for readers to better understand the article. 

The abstract section is a bit long. The findings obtained from the study can be presented in a more summarised form. 

Current sources were used in the literature review in the study. The process is well explained. There is no disconnection in the subject transitions between paragraphs.

2.2.1. Data sources, The resolution of the DEM map used as data is 12.5 metres. Sentinel-2 multispectral RS image data resolution is 10 metres. It can be stated whether the resolution of the data used in this section is sufficient or not. In addition, it should be explained how the results of the study may change in case of higher resolution data. This will guide the readers and scientists who will work on the subject.

Section 2.3, Methods, explains why the Kruskal algorithm was used. There are many more methods.

Figure 2. Technical route. Is it appropriate to change it to the flowchart in this study?

3.1.3. Land use type structure of ecological zone, In the conservation priority area, the proportion of 267 the area of water body, wetland, forest and grass land is significantly higher than other 268 types of land, which is 11.78%, 9.43%, 8.46% and 6.41% respectively.

It is not clear from the article how these values were found. Were these data obtained by classifying the remote sensing data? Which methods were used? The accuracy of the data obtained should be added. 

The conclusions section is well designed.

Reviewer 4 Report

Comments and Suggestions for Authors

Dear authors,    
you have written an interesting article according to my opinion and to my field of interest.

The theme of the paper falls within the scope of the journal's topics, making the research potentially interesting to a relatively narrow circle of experts. However, to position the study correctly in an international context, it is necessary to elaborate on the methodology and explain the main research questions in more detail. It's not entirely clear what the smallest units of land consolidation are and how priorities for ranking them are determined. The paper applies methods that are already established and common in practice, which are also mentioned and explained in numerous works, even within this journal. What is the novelty of the applied method, and do its application yield different results from those currently in common practice? 

Besides combining several common ranking methods, I did not notice any novelty that, although I fully understand the amount of professional work required for such a project, would be suitable for publication in a scientific journal. The conclusions and results are overly generic, and apart from a visual comparison with the existing land consolidation plan, there is no detailed explanation of the results or a critical review of the data determined by the model.

Therefore, I must conclude that the paper, in its current form, is not suitable for publication and requires major revision before consideration for publication.

In addition to the general comments above, please also consider the following specific parts of the paper: 

- the term "comprehensive land consolidation" is mentioned in the title and later in the introduction without explanation or clarification of its necessity

- what would be the difference if another land consolidation method were applied, especially since there is no detailed spatial/physical planning within the consolidation area? 

-the introduction should summarize the need for research and provide an overview of previous research in the same area. In the paper, the introduction mixes and explains part of the methodology. I suggest moving this content to Chapter 2, Methodology

- the Methodology section contains almost everything except a detailed explanation of the methodology. Section 2.3 contains only two sentences explaining a rather extensive image. Figure 2, on the other hand, shows a relatively complicated flowchart that is not justified further in the text. 

- Chapter 2.3.1 seems quite unclearly written, and it is not explained in detail how the spatial units mentioned were created. 

- explain why the weights are determined as stated in the paper, and what differences would arise if the weights were distributed differently. 

- Figure 7 is not clear enough - a relatively large number of different zones with similar shades make it difficult to understand the space; I suggest highlighting only those classes important for consolidation to make the situation clearer to readers.

- what is the difference between figures 4.a and 8? They seem almost identical. Do both need to be included in the paper? 

- the conclusion must be more extensive: what conclusions are drawn from the research, how does this solution differ, or why doesn't it differ from current policy? Is such extensive research necessary if existing planning yields similar results? What is the cost-benefit of the applied method, and how can the results of the paper be placed in an international scientific context through the potential application of the results to other regions of the world?

Reviewer 5 Report

Comments and Suggestions for Authors The work is thoroughly prepared from the workshop side. The methodology of the study was well described and the discussion was conducted in great detail. The graphic material is of a high standard. For me, the conclusions are too general and should be improved.    

Round 2

Reviewer 3 Report

Comments and Suggestions for Authors

I would like to thank the authors for making the changes in full. After the changes, the article is now easy for readers to read and understand. The coherence of meaning between paragraphs has also been restored. The flow chart summarising the study is also much more understandable. The conclusions obtained in the results section are summarised with references to tables and pictures. The article now looks better from a scientific point of view.

Reviewer 4 Report

Comments and Suggestions for Authors

Dear Authors, I have reviewed the revisions - most of the comments have been addressed sufficiently. I believe the manuscript has been sufficiently improved to warrant publication in Land.